# Identification of Anthocyanins and Their Fouling Mechanisms during Non-Thermal Nanofiltration of Blueberry Aqueous Extracts

**DOI:** 10.3390/membranes11030200

**Published:** 2021-03-12

**Authors:** Ming Cai, Chunfang Xie, Huazhao Zhong, Baoming Tian, Kai Yang

**Affiliations:** Department of Food Science and Technology, Zhejiang University of Technology, Hangzhou 310014, China; x987665926@163.com (C.X.); 2111926019@zjut.edu.cn (H.Z.); tbm2020@zjut.edu.cn (B.T.); yangkai@zjut.edu.cn (K.Y.)

**Keywords:** anthocyanins, nanofiltration, fouling mechanisms, interactions, membrane

## Abstract

Organic fouling in the nanofiltration (NF) process, which is a non-thermal technology to recover active components, is a critical problem limiting its applications. This study seeks to identify the anthocyanins on the NF membrane and explore their fouling mechanisms during concentration of blueberry extracts. Seven kinds of monomeric anthocyanins in foulants—delphinidin-3-O-galactoside, delphinidin-3-O-glucoside, delphinidin-3-O-arabinoside, cyanidin-3-O-galactoside, petunidin-3-O-galactoside, peonidin-3-O-glucoside, and malvidin-3-O-glucoside—were identified. Moreover, chalcone, myricetin derivative, and an unknown substance with [M^+^H]^+^ at m/z 261.1309, which is the fragment ion corresponding to the break of glycoside bond of anthocyanins, were obtained. Interactions between anthocyanins and membrane made from polyamide were principally governed by the CH-π and π-π stacking of aromatic rings, the establishment of hydrogen bonds, and electrostatic interaction. This study will be helpful to further control fouling and choice of cleaning agents in concentration of anthocyanins-rich extracts.

## 1. Introduction

Blueberry (*Vaccinium corymbosum*) and blueberry extracts are known for their health benefits due to a high content of anthocyanins. As a natural pigment, anthocyanins are a potential substitute for synthetic dyes due to their safety and special aroma. In addition to being used as colorants, anthocyanins with high antioxidant capacity play important roles in prevention of many degenerative diseases, such as cancer, cardiovascular issues, and Alzheimer’s by providing neutralization of unstable free radicals [1]. In addition to being used in the fields of functional food, medicine, and cosmetics that we are familiar with, anthocyanins and other biophenols have also been added in smart nanofilms for food packaging due to its pH sensitivity in recent years [2], or embedded in nanoparticles for milk freshness detection [3] and other new potential applications fields, such as polymer dots [4], membranes [5] and fuel cells [6]. The growing interest in the recovery of anthocyanins and their utilization has urged researchers to develop environmentally friendly extraction and purification procedures in order to preserve their stability. Concentrated juice or extracts are the most common preservation methods, providing a number of benefits, such as reduced volume or weight, reduced packaging, easier handling and transportation, and extended shelf life. Some thermal concentrate technologies are widely used in the food industry. However, these techniques result in color changes, loss of bioactive compounds, and reduction in antioxidant activity of the products [7,8].

Nanofiltration (NF), as a non-thermal technology, replaces some traditional concentration methods because of its advantages of low energy consumption, reduction in processing steps, greater separation efficiency, and assisted extraction for the recovery of heat-sensitive substances such as biophenols [9,10,11,12,13]. However, membrane fouling is a bottleneck limiting its engineering applications. Blueberries contain a large amount of phenolics, especially anthocyanins, which tend to interact with other components, such as protein, carbohydrate, lipid, and other secondary metabolites existing in the juice or concentrates, especially during concentrate processing [14]. The formation of complexes or micelles on the membrane as organic pollution increases the difficulty in extracting and identifying anthocyanins from the polymer membrane. However, very few studies have reported on this issue. 

To characterize and identify foulants, one of the methods used is to first remove the compounds from the membrane. Nevertheless, extraction of organic foulants from membrane materials, especially unstable phenolic compounds, has been rarely reported [15,16]. However, in terms of anthocyanins, it was only detected in a fouled membrane carried out with a model solution [15]. Studies on extraction and identification of organic foulants from fouled membranes are scarce, especially unstable compounds such as anthocyanins. Polyphenols are amphipathic molecules, with hydrophobic aromatic rings and hydrophilic (acidic) phenolic hydroxyl groups. Their adsorption on polymeric membranes involves both hydrophobic effects and the formation of hydrogen bonds [17]. It is noteworthy that polyphenols may interact with other compounds to form large particles, which have a negative impact on the filtration process. These organic contaminations increase during the continuous operation, which could change the internal structure of the membrane, causing irreversible fouling, shortening of membrane service life, and increasing operating costs [18]. The initial decline in permeate flux values is visualized in the first 20 min of the NF process for concentration of phenolic compounds from strawberry; it is related to the aggregation of phenolic compounds and polysaccharides in the pore entrance and on the membrane surface. Moreover, their interactions and those with the membrane surface and pores play a key role in membrane fouling [16]. Hence, it is necessary to understand the complex interactions between the foulants and the membrane. To our knowledge, there are no studies on the effects of anthocyanins as membrane fouling and their interaction with membranes during the concentration process. 

The objectives of this study were to extract and identify the anthocyanins on/in the fouled NF membrane after concentration of blueberry extracts and to demonstrate the fouling mechanisms between anthocyanins and membrane material.

## 2. Materials and Methods

### 2.1. Materials and Reagents

Dried blueberries were purchased from Xing’ an Wild Blueberry Shop (Heilongjiang, China), and stored at 4 °C for further experiments. Methanol and acetonitrile of chromatographic grade, potassium chloride (>99.99%), sulfuric acid (98%), muriatic acid (38%), and formic acid (chromatographic grade) were purchased from Aladdin (Shanghai, China). 

Commercial NF flat sheet membrane made from polyamide with molecular weight cut-off (MWCO) ranging from 500 to 1000 Da was purchased from RisingSun Membrane Technology Co., Ltd (Beijing, China) and used to concentrate blueberry extracts. 

### 2.2. Preparation of Blueberry Extracts

Blueberry extracts were obtained independently according to a described method [19] with some modifications. Briefly, 25 g blueberries crushed with a food processor (JYL-C022E, Joyoung, Hangzhou, China) were dispersed in 2 L pure water in a glass bottle on a magnetic stirrer (HJ-4, YUHUA, Gongyi, China) at a speed of 750 rpm in 25 °C for 1 h in the dark. The aqueous extracts were then centrifuged and stored at −18 °C in an airtight amber glass flask not exceed 24 h for further NF.

### 2.3. NF Concentration of Blueberry Extracts

The NF processes were carried out as shown in Figure 1. NF membranes were conditioned in distilled water at room temperature for 1 h prior to the process. This pre-treatment improves the permeate flux, ensuring that the membrane is completely wet by the solvent and removes the residual protein or other impurities on the membrane surface. Then, the membrane was loaded onto the membrane module, and prepressed with deionized water for 30 min. Finally, the conditions of membrane filtering water at a pressure close to the one used for concentration was performed.

1 L blueberry extract was concentrated at a transmembrane pressure of 1.0 MPa, feed flow at 0.08 L/min, active membrane area of 7.34 × 10^−3^ m^2^, and temperature of 25 ± 2 °C. Concentrates of blueberry extracts with a volume reduction factor (VRF) of 2.0 were obtained. Feed and concentrate were collected for further analysis. The fouled membrane after NF was placed in 25 ± 2 °C and dried away from light for 24 h for further determination. 

### 2.4. Characterization of Membrane and Foulants

#### 2.4.1. Permeate Flux

To evaluate the performance of NF membranes, permeate flux *J*_v_, and volume concentration ratio *VRF* were calculated as per the following equations [20]:(1)$$J_{v}=\Delta V\;\times \;{\left(\;A_{m}t\right)}^{-1}$$
(2)$$VRF=Vo\;\times \;{V_{f}}^{-1}$$
where *J*_v_ is the permeate flux during NF (L (m^2^ h)^−1^); Δ*V*, permeate volume (L) collected at the same interval *t* (h); *A*_m_, active area of membrane of 7.34 × 10^−3^ m^2^; *V*_o_ and *V*_f_, initial and final volume (L) of the feed and retentate, respectively.

#### 2.4.2. Atomic Force Microscopy (AFM)

Morphology of NF membranes before and after the process was identified by AFM, according to a reported method [21]. The dried membrane, approximately 1 cm × 1 cm, were fixed on a sample holder and scanned in tapping mode (semi contact). AFM imaging was conducted on 90 μm × 90 μm sample areas. Root mean squared roughness (Rq) of membranes was determined from 25 μm × 25 μm height images using the Nanoscope Analysis 1.8 (Dimension Icon, Bruker, Billerica, MA, USA).

#### 2.4.3. Contact Angle

Contact angle was performed in static mode on a tensiometer (OCA15EC, Dataphysics, Germany). Pristine membrane and fouled membrane were tested for their affinity to water at 25 °C, respectively. Membrane samples (0.05 m × 0.02 m) were set on a glass plate, and 5 μL water were carefully dropped onto their surface. Drop images were taken automatically as a function of time.

#### 2.4.4. Fourier-Transform Infrared (FTIR)

FTIR spectroscopy was carried out using a Nicolet IS10 spectrometer (Thermo-Scientific, Madison, WI, USA) equipped with a Globar source and a DTGS detector from 400 to 4000 cm^−1^. All spectra were collected at the average of 32 scans with a resolution of 8 cm^−1^. 

#### 2.4.5. Extraction of Fouling Compounds

The fouled membranes were cut into small pieces to increase the contact surface for solvent extraction. Extraction solvents were combined as per a previously reported method [15] with some modifications. Briefly, 0.4 g fouled membrane fragments were transferred to an amber flask with 10 mL acetonitrile/methanol/isopropanol/water/acetic acid (25:25:25:25:0.1, *v*/*v*/*v*/*v*, MeCN/MeOH/H_2_O/HOAc). The flask was sealed and placed in a magnetic stirrer (HJ-4, YUHUA, China) operating at a stirring speed of 750 rpm and 25 °C for 1.5 h in darkness. The mixtures were then centrifuged at 5000 rpm for 10 min at 25 °C to remove the residue, and the supernatant was recovered for anthocyanins analysis.

#### 2.4.6. HPLC Analysis

Anthocyanins analysis was carried out according to HPLC methods [22] with some modifications. The samples were filtered through a 0.22 μm polyvinylidene fluoride membrane before HPLC (Waters 2695, Milford, MA, USA) with a reverse phase C18 column (ZORBAX C18 250 mm × 4.6 mm × 5 μm, Agilent, Santa Clara, CA, USA) and wavelength of 520 nm. Mobile phase A (5% aqueous formic acid) and B (MeCN) with a flow rate of 1 mL/min were used. A gradient program was set as follows: 5% B (0 to 5 min), 5% to 10% B (5 to 15 min), 10% B (15 to 28 min), 10% to 13% B (28 to 35 min), 13% to 15% B (35 to 50 min), and 5% B (50 to 55 min). The column was maintained at a constant temperature of 25 °C, and injection volume was 10 µL. 

#### 2.4.7. HPLC-ESI-MS Analysis

Anthocyanins were identified by HPLC-ESI-MS using an Agilent 1100 HPLC equipped with a UV detector and LCQ ion-trap mass spectrometer (MS) fitted with an electrospray ionization interface (ESI) (Waters UPLC-Synapt G2, Milford, MA, USA). ESI capillary voltage was 3.0 kV in positive-ion mode with capillary temperature of 325 °C. A nebulizing gas of 10 L min^−1^ and drying gas of 10 L min^−1^ were applied for ionization using nitrogen. ESI was performed with a scan range between 100 and 1200 m/z.

#### 2.4.8. Determination of Monomeric and Polymeric Anthocyanins

Total monomeric anthocyanin content (TAC) was determined according to the spectrophotometric pH differential method [23]. Aliquots were diluted with potassium chloride buffer (pH 1.0) and sodium acetate buffer (pH 4.5). These samples were thoroughly homogenized and placed in the dark for 30 min to equilibrate. The absorbance of supernatant of the fouled membrane extracts was measured using a spectrophotometer (UV2450, Shimadzu, Kyoto, Japan) at 520 nm and 700 nm, respectively. *TAC*_1_ was calculated using the following equation:(3)$$TAC_{1}\left(\frac{mg}{mL}\right)=\left(\left[{\left(A_{520}-A_{700}\right)}_{pH1.0}-{\left(A_{520}-A_{700}\right)}_{pH4.5}\right]\times MW\times DF\right)\;\times \;{\left(\varepsilon \times L\right)}^{-1}$$
where *M*_W_ is the molecular weight of the anthocyanins (cyanidin-3-O-glycoside, 449.2 g mol^−1^); DF, dilution factor; *ε*, extinction coefficient (26,900 L (mol cm)^−1^); and *L*, path length (1 cm).

Total monomeric anthocyanins from the fouled membrane were calculated by the following formula:(4)$$TAC_{2}(\mathrm{mg}/g)=TAC_{1}\times v\times m^{-1}$$
where *v* is the solvent volume for extraction of membrane fouling, mL; *m* is the quality of the contaminated membrane, g.

Percentage of polymeric anthocyanins was determined using a method described in [24]. The extracts were diluted with water to make an absorbance between 0.5 and 1.0 at 520 nm. For analysis, 0.2 mL of 0.9 M potassium metabisulfite was added to 2.8 mL of the diluted sample, and 0.2 mL distilled water was added to 2.8 mL of the diluted sample as the control. After equilibration for 15 min, the samples were evaluated at λ = 700, 520, and 420 nm, respectively. Color density (K) was calculated using the control sample according to the following formula:(5)$$K=[\left(A_{420nm}-A_{700nm})+\left(A_{520nm}-A_{700nm}\right)\right]\times DF$$

Polymeric color (P) was determined using the bisulfite bleached sample as the following formula:(6)$$P=\left[\left(A_{420nm}-A_{700nm}\right)+\left(A_{520nm}-A_{700nm}\right)\right]\times DF$$

Total polymeric anthocyanins in the fouled membrane were calculated by the following formula:(7)$$polymeric\;anthocyanins\;\left(mgb/g\right)=\frac{TAC_{2}}{P-K}\times P$$

There were no significant differences between the determinations in the extracts and in the fouled membrane; all experiments were conducted three times in parallel.

### 2.5. Statistical Analysis 

All data presented are the mean value ± standard deviation of three independent experiments. Figures were obtained using GraphPad Prism Version 7 (GraphPad Software, Inc., Chicago, IL, USA). Differences were considered significant at *p* < 0.05.

## 3. Results and Discussion

### 3.1. Characterization of Membrane Fouling in the NF Process

The permeate flux of NF with blueberry extracts is displayed in Figure 2a. Flux performance was similar to a typical flux curve of permeation systems composed of porous membranes, in which it decreased due to concentration polarization and adsorption of solutes onto the membrane. As reported, accumulation of anthocyanins during the process generated more interactions with simple sugars and polysaccharides, or homologous species such as other anthocyanins or phenolic acids for polymerization or aggregation into colloidal particles deposited on the membrane surface [14]. This phenomenon results in pore occlusion, which further leads to membrane incrustation and gradual flux decrease.

The pH value of the blueberry extract is about 3.1. At this value, anthocyanins exist in the form of red flavonoid cations, which are water-soluble pigments with small molecules. They can be adsorbed on the membrane surface or penetrate into the membrane, which also confirms the red fouling on the surface in Figure 2—it could have been caused by the interaction between anthocyanins and other organic components in the extraction system.

Morphology of membranes before and after NF treatment is shown in Figure 2b,c. Roughness of the pristine NF membrane was 10.6 nm, which increased to 47.8 nm after concentration. The reason for this increase in surface roughness was the deposition of a thin cake layer on the membrane, which was formed by the aggregation of extracts during the NF process. On the other hand, the roughness also depends on the wettability of the membrane surface, which is directly related to the contact angle of the membrane. It has been reported that hydrophilicity decreased when the roughness increased [25]. The increase in contact angle (as shown in Figure 3), hydrophobicity, and roughness of contaminated membrane were also found to be corresponding. 

### 3.2. Characterization of Anthocyanins in the Fouled Membrane

#### 3.2.1. Content of Anthocyanins on the Fouled Membrane

As shown in Table 1, TAC and polymeric anthocyanins were about 0.27 mg g^−1^, and 0.72 mg g^−1^ in the fouled membrane, respectively. After NF, the monomeric and polymeric anthocyanins of the concentration increased to 1.8 and 2.7 times, respectively, where the content of polymeric anthocyanins was found to be greater than that of the monomeric anthocyanins. This result can be interpreted as a result of anthocyanins self-associating or interacting with other components in the aqueous extract system to form polymer accumulation on the membrane surface during the concentration.

#### 3.2.2. Identification of Anthocyanins in Foulants

Fourteen different monomeric anthocyanins were detected in blueberry extracts in our study; the HPLC chromatograms are presented in Figure 4a, and peak identification is summarized in Table 2. As shown in Figure 4b, seven kinds of anthocyanins, delphinidin-3-O-galactoside, delphinidin-3-O-glucoside, delphinidin-3-O-arabinoside, cyanidin-3-O-galactoside, petunidin-3-O-galactoside, peonidin-3-O-glucoside, and malvidin-3-O-glucoside were determined in the foulants of the membrane after NF, which were the same as those in the extracts. Many factors could cause anthocyanins deposition on the membrane, such as membrane material, membrane surface polarity, molecular weight of solutes, and interactions between the solute and the membrane [26]. It indicated that membrane surface polarity is a major determinant in building of polyphenol deposit (amounts, nature of the deposited molecules and reversibility of the deposit). A membrane is constructed with polyamide networks containing aromatic moieties and these structures have non-beneficial surface affinity towards the rejected polyphenols, possibly leading to enhanced adsorption of polyphenols [23]. In addition, retention characteristics were affected by the relative position of the substituted groups in the aromatic ring. Typically the retention increased following the sequence of para→meta→ortho. This is because the shift of the second substituted functional group from para to meta and ortho positions in the benzene ring promotes a slight increase in molecular size. This effect is attributed to the attraction exerted by the substituted groups on the aromatic electronic cloud with a consequent modification of carbon interatomic distances [27]. Therefore, for polyhydric phenols, steric hindrance is recognized as the main factor affecting membrane retention.

It is interesting that there are three additional peaks, (1), (2), and (3), which differ from the original anthocyanins in the extracts, as shown in Figure 4b. This indicates that peak (1) gives an [M^+^H]^+^ ion at m/z 261.1309; this fragment ion might correspond to the break of the glycoside bond of anthocyanins. The ion at m/z 323 was also observed as one of the main ions due to losses of water molecules, cleavages of a pyranic sugar ring which implies breakage of the C1"bond, and the loss of the glucosidic methylol group as formaldehyde ([M^+^H–3H_2_O–2CO]^+^), as previously reported [28]. Peak (2) was preliminarily identified as chalcone according to its MS measurement of [M^+^H]^+^ ion at m/z 305.1603 (C_14_H_25_O_7_, 1.0 ppm), shown as Figure 4f; this corresponds to anthocyanins existing as highly stable flavylium cations with a bright red colour. It changes to a colourless carbinol pseudo-base upon hydrolysis in aqueous solutions, then undergoes ring opening to form a yellow chalcone, and finally decomposes into 2,4,6-trihydroxybenzaldehyde and p-hydroxybenzoic acid or their derivatives [29]. Similarly, as displayed in Figure 4d, peak (3) was putatively identified as a myricetin derivative according to a published study [30], as its fragment ion at m/z 319, which represented the loss of glucuronic acid moiety. Moreover, this myricetin derivative has been previously reported in blueberry pomace extract [23].

### 3.3. Fouling Mechanisms between Anthocyanins and Membrane

#### 3.3.1. Changes in Hydrophobic Characteristics of the Membrane Surface

Figure 3 shows that the original NF membrane was hydrophilic, as contact angle was <90°. It can be interpreted as: the polymeric main chain of a polyamide membrane contains a polar amide group, which can interact with a polar solvent through hydrogen bonds and thus presents a low contact angle [31]. Besides, there are fouling molecules, such as polyphenols, tannins, polysaccharides, and proteins, in blueberry extracts, where anthocyanins are amphiphilic molecules with hydrophobic aromatic rings and hydrophilic phenolic hydroxyl groups [32]. The hydrophilic groups were easily adsorbed on the membrane or the membrane surface, while the hydrophobic groups were exposed on the membrane surface, which increased the hydrophobicity of the membrane [33]. Therefore, the membrane contact angle increased from the initial value of 30.35° ± 0.32° to 42.3° ± 0.41° after concentration.

#### 3.3.2. Interaction between Anthocyanins and Membrane

In Figure 5, FTIR spectra of the membrane surface before and after NF provides additional information about the chemical nature of foulants and their interactions. The strength of the infrared absorption band can be used to quantify the amount of target compounds deposited on the membrane [15]. According to previous studies, polysaccharides usually have a strong C-O stretching band between 1000 and 1040 cm^−1^; the peak at 1035 cm^−1^ indicates the accumulation of polysaccharides on the membrane surface. The peak at 1217 cm^−1^ is due to the accumulation of phenolic acid [34]. According to some previous studies, the peak of polyphenols is around 1393 cm^−1^. Moreover, peaks located at 1440, 1523, and 1622 cm^−1^ in the C=C spectral region (1450–1650 cm^−1^) can be interpreted as enlargement and intensification of bands by accumulation of aromatic rings of anthocyanins. There also is a blue-shift of bands to higher wavenumbers under π-π and CH-π interactions [35]. Moreover, anthocyanins are amphipathic molecules with hydrophobic aromatic rings and hydrophilic (acidic) phenolic hydroxyl groups, and anthocyanins exist as positively charged flavone cations. NF membranes made from polyamides produced more carboxyl negative charges and amino positive charges on the membrane surface under acidic conditions [36]. Thus, their adsorption on polymeric membranes involves two hydrophobic effects—the formation of hydrogen bonds and electrostatic interaction [17].

The stretching vibrations band of -OH bonds at 3400 cm^−1^, while characteristic of bands in the FTIR spectra of the fouled membrane, saw a red-shift of the band to lower σ by 50 cm^−1^ with hydrogen bonds created by links between water in the polymer matrix and O in the phenolic compounds [35].

Though the rejection in membrane is mainly focused on the MWCO of the membrane, the interactions between membrane materials and feed components play a significant role in offering selectivity. By comparing the spectra and the assignments of characteristic bands in the pristine and fouled membranes, they can be summarized as changes in spectra caused by different types of interactions between the membrane and anthocyanins due to organic fouling, especially CH-π and π-π interactions, hydrogen bonds and electrostatic interaction. Similar to the results obtained by Huang et al. [37], hydrogen bonds, hydrophobic interactions, and benzene ring interactions via π-π stacking were considered as the main interactions involved in the adsorption of phenolic compounds onto polyethersulfone membranes. Similar interactions were also observed in the case of fouling analysis of porous membranes containing electrodialysis, polyvinyl chloride, or polysulfone for the treatment of peptide solutions or lignocellulosic biomass hydrolysates that include humic substances, polysaccharides, sugars, phenolic compounds, etc. 

## 4. Conclusions

After NF concentration, the deposition of anthocyanins and other substances on the surface of the NF membrane made the contact angle increase from 30.35° ± 0.32° to 42.3° ± 0.41°, and the roughness from 10.6 nm to 47.8 nm. The TAC and polymeric anthocyanins were found to be about 0.27 and 0.72 mg·g^−1^ in the fouled membrane, respectively. Seven kinds of monomeric anthocyanins, chalcone, and myricetin derivatives were successfully extracted and identified from the fouled NF membrane after concentration of blueberry extracts. It showed the presence of interactions between anthocyanins and membranes by comparison and interpretation of FTIR spectra of the pristine and fouled membranes. Enlargements and blue shifts of some bands were observed under the action of CH-π and π-π interactions of aromatic rings in anthocyanins and aromatic rings of the polymer matrix or due to establishing hydrogen bonds. Moreover, it was clear that deposition of these compounds on the membrane surface increased the contact angle and roughness of the membrane. These results will be helpful in the development of adequate or new strategies for the prevention and control of fouling, and in selecting specific agents in cleaning solutions for the NF membrane. 

## Figures and Tables

**Figure 1 membranes-11-00200-f001:**
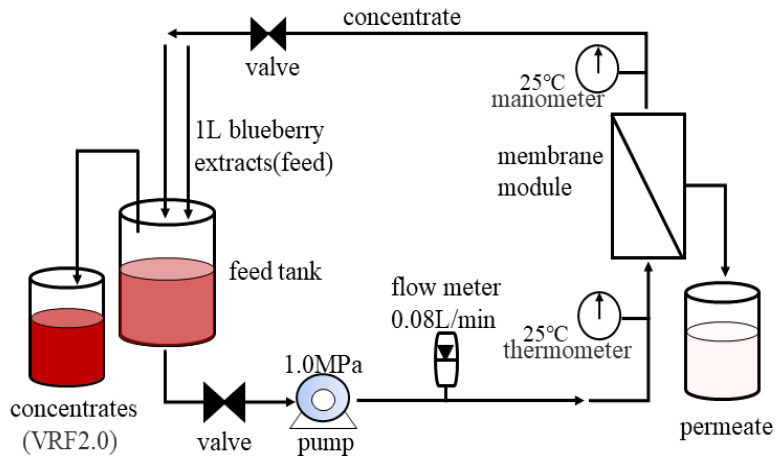
Schematic diagram of the NF system.

**Figure 2 membranes-11-00200-f002:**
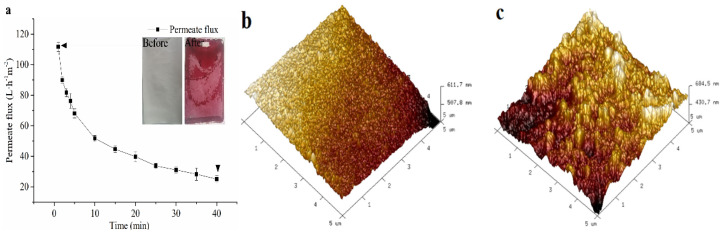
Changes of flux and membrane morphology during NF of blueberry extracts. (**a**) permeate flux; (**b**) pristine membrane; (**c**) fouled membrane.

**Figure 3 membranes-11-00200-f003:**
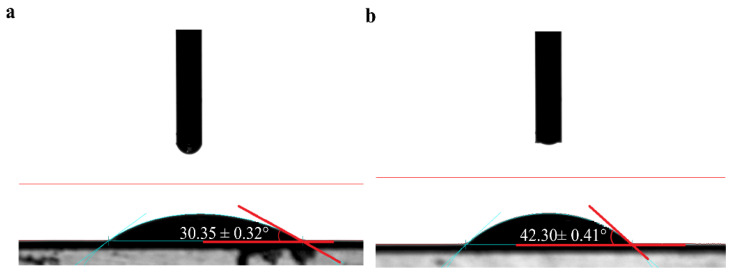
Change in contact angle. (**a**) pristine membrane; (**b**) fouled membrane.

**Figure 4 membranes-11-00200-f004:**
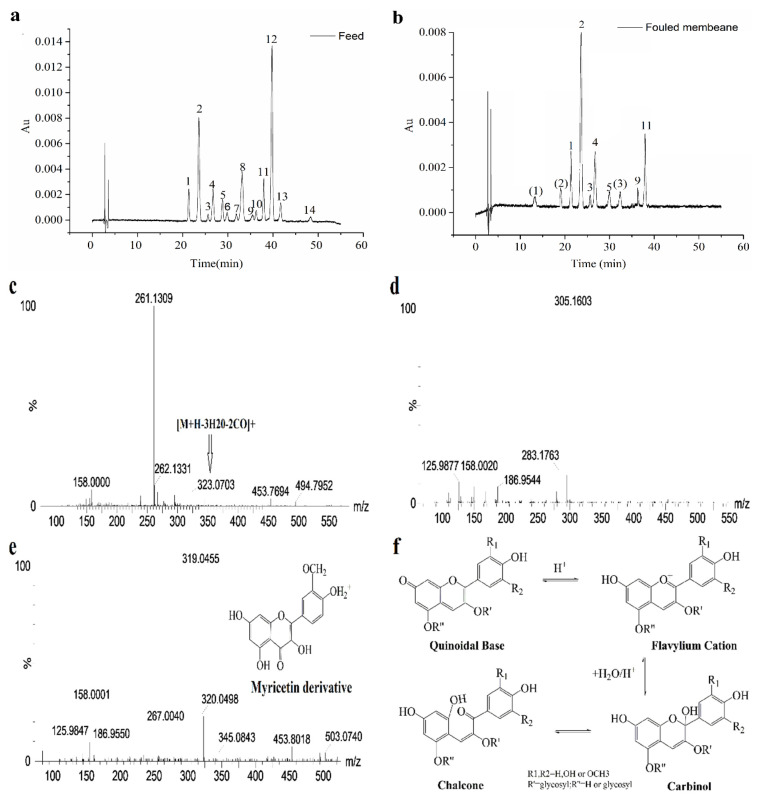
HPLC chromatogram profiles of anthocyanins in extracts and foulants. (**a**) anthocyanins of feed; (**b**) anthocyanins in membrane foulants (peaks’ assignments are shown in Table 2); (**c**) MS spectra of peak (1); (**d**) MS spectra of peak (2); (**e**) MS spectra of peak (3); (**f**) Equilibrium between the four forms of anthocyanins [29].

**Figure 5 membranes-11-00200-f005:**
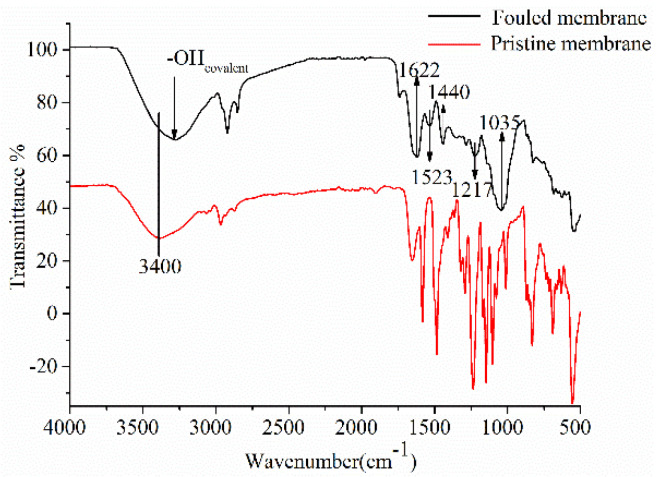
FTIR spectra of pristine and fouled membranes.

**Table 1 membranes-11-00200-t001:** Content of anthocyanins in membrane foulants and feed.

	TAC	Polymeric Anthocyanins
Foulants in membrane (mg/g)	0.27 ± 0.03 ^b^	0.72 ± 0.01 ^a^
Feed (mg/L)	90.5 ± 0.14 ^d^	27.81 ± 0.04 ^c^
Concentrate (mg/L)	167.32 ± 0.22 ^a^	76.05 ± 0.00 ^a^

^a–d^ Values in the same column with different letters are significantly different (*p* < 0.05).

**Table 2 membranes-11-00200-t002:** Individual anthocyanins and their derivates in extracts and foulants. Identification based on MS and published data [22].

Peak No.	tR(min)	MS/MS2	Identification
1	18.00	465/303	delphinidin-3-O-galactoside
2	20.45	465/303	delphinidin-3-O-glucoside
3	23.10	435/303	delphinidin-3-O-arabinoside
4	24.78	449/287	cyanidin-3-O-galactoside
5	25.65	479/317	petunidin-3-O-galactoside
6	27.53	479/317	petunidin-3-O-glucoside
7	29.67	449/317	petunidin-3-O-arabinoside
8	31.21	493/331	malvidin-3-O-galactoside
9	31.21	463/301	peonidin-3-O-glucoside
10	33.06	493/331	malvidin-3-O-glucoside
11	34.87	463/331	malvidin-3-O-glucoside
12	37.04	465/303	delphinidin-3-O-xyloside
13	38.14	479/303	delphinidin -3-O glucoside acid
14	43.41	463/303	delphinidin-3-O-rutin
(2)	16.77	305	chalcone
(3)	30.06	319/273	myricetin derivative

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
