# Peer review of "Identification of Anthocyanins and Their Fouling Mechanisms during Non-Thermal Nanofiltration of Blueberry Aqueous Extracts"

_membranes, 2021, doi:10.3390/membranes11030200_

Round 1
Reviewer 1 Report
The research by Cai concerns the fouling of nanofiltration membranes using blueberry extracts. The work is novel and it is interesting for the readers of Membranes. The manuscript has detailed analysis, and the conclusions are supported by the results. The following points need to be addressed.
1. The process schematic (Figure 1) should have all the process parameters and their ranges to aid understanding, e.g. concentrations, temperature, flow rates, pressure etc.
2. The derivation of errors reported in the manuscript should be provided. How many measurements were carried out? Were independently prepared samples used?
3. Nanofiltration assisted extraction of plants should be mentioned (10.1039/C7GC00912G; 10.1016/j.jclepro.2020.123349).
4. The inset in Figure 4b is not visible. It should be included as a separate panel enlarged to an appropriate size.
5. All the structures provided for the MS analysis are distorted, bonds are missing (low resolution) and have different sizes. Provide the structures in a palatable way.
6. There are multiple options for the interaction points between delphinidin and polyamide. The representation in Figure 6 is speculative and evidence should be provided that interactions occur as the authors depicted. If there is no direct evidence then delete the figure.
7. Line 377: volume and page numbers are missing; line 408: name 'Y. J. J. o. F. Q.' incorrect.
Author Response
Dear Editor,
Thank you and the referee for reviewing our revised manuscript (ID: membranes-1134930) carefully. For the comments of reviewer 1#, we discussed and thought about it thoroughly. Please see the details as the following:
- The process schematic (Figure 1) should have all the process parameters and their ranges to aid understanding, e.g. concentrations, temperature, flow rates, pressure etc.
ANS: Thank you for your suggestion, the process schematic (Figure 1) has been improved.
- The derivation of errors reported in the manuscript should be provided. How many measurements were carried out? Were independently prepared samples used?
ANS: Thank you for your comments. The derivation of errors has been added in the manuscript, and information of the experiments were described in Section 2.5.
- Nanofiltration assisted extraction of plants should be mentioned (10.1039/C7GC00912G; 10.1016/j.jclepro.2020.123349).
ANS: Thank you for your suggestion. These literatures have been cited as the references.
- The inset in Figure 4b is not visible. It should be included as a separate panel enlarged to an appropriate size.
ANS: Thank you for your comment. It has been revised as shown in Figure 4.
- All the structures provided for the MS analysis are distorted, bonds are missing (low resolution) and have different sizes. Provide the structures in a palatable way.
ANS: Thank you for your comment. It has been revised as shown in Figure. 4.
- There are multiple options for the interaction points between delphinidin and polyamide. The representation in Figure 6 is speculative and evidence should be provided that interactions occur as the authors depicted. If there is no direct evidence then delete the figure.
ANS: Thank you for your suggestion. Currently, there is no evidence obtained in our study, accordingly, we think it is better to delete the figure in the manuscript.
- Line 377: volume and page numbers are missing; line 408: name 'Y. J. J. o. F. Q.' incorrect.
ANS: Thank you for your comment. Sorry for our carless, the volume and page numbers have been added, and the name has been revised.
Please find the revised manuscript in which all the revised terms have been highlighted.
If there is any problem in our current study and manuscript, please feel free to contact me. We will consider it more seriously.
Regards,
All authors

Reviewer 2 Report
The authors have improved the manuscript, clarifying most of the issues stated in the first revision. However, there are still certain comments that have not been answered satisfactorily. Please find below the answers to the comments that need to be re-revised.
Comments of reviewer 3
Lines 58-60:
The authors refer to membrane fouling during wine clarification, which is most probably carried out by microfiltration. Fouling mechanisms for nanofiltration should be referred. It is true that even though the size of the molecules is much smaller than the microfiltration pores, fouling occurs due to the interactions between these molecules (polyphenols) and other molecules present. But there is a scarce review on polyphenol fouling during nanofiltration in the introduction.
ANS: Thank you for your suggestion. Some literatures of fouling mechanisms in nanofiltration have been cited in our manuscript.
The authors have improved the introduction, adding relevant information regarding previous studies related to their work. In lines 53-54 of the revised manuscript, they have maintained a sentence regarding the need of extracting the foulants for membrane fouling characterization. This is an option, but not the only one. In fact, most of the membrane fouling characterization has been performed based on characterization of foulants deposited on the membrane. The authors should clearly state that extraction is one possibility, but not the only one.
Line 109: “different membranes were tested for their affinity of water at 25ºC” not clear what the authors mean.
ANS: Thank you for your comment. “Different membranes were tested for their affinity of water at 25ºC” is to measure the change of hydrophilicity and hydrophobicity of the membrane. It has been improved in the manuscript.
This is not still clear from my point of view. Do the authors mean that several samples of the NF membrane were tested, or different membranes refer to membranes other than the one employed in the experiments? I think the word “different” can be misleading.
In Figure 3b it is not possible to clearly read the additions in the figure.
ANS: Thank you for your comment. The chromatogram has been modified in the manuscript.
Still the quality of the figure could be improved
Lines 144-159: it is not clear how they perform the determination of the TAC2. First, they obtain the mg/mL of monomeric anthocyanins in the extracts from the membrane (TAC1). Then, they calculate the monomeric anthocyanins for the fouled membrane, using the previous value in mg/mL, which is multiplied by a dilution factor (that was already used in the TAC1) and divide by the weight of membrane (quantity not quality). Therefore, the final result is referred to the g of membrane. Why do they use these units? Would not be better to refer to membrane surface unit? Also, in this part it should be clarified if there is any difference between the determinations in the extracts and in the fouled membrane.
ANS: Thank you for your comment. Total monomeric anthocyanin content was determined according to the method described in [21]. Firstly, mixed organic solvents were used to extract anthocyanins from 0.4g dry fouled membrane fragments, and the anthocyanin content((TAC1) in the extract solvents was determined. Then, TAC1 was used to calculate the anthocyanins content (TAC2) in unit weight of contaminated membrane. Because of the uneven distribution of membrane fouling, each representative dry membrane was selected to determine its total anthocyanin content by weight. Therefore, the final result is referred to the g of membrane. There is no significant difference between the determinations in the extracts and in the fouled membrane.
Thank you for the clarification. I have no problem regarding the method describes in reference [21]. It could be added in the manuscript that the mg/L for TAC1 are referred to cyanidin-3-O-glycoside. My concern is for the way TAC2 is calculated, I see that it has to be multiplied for the volume used in the extraction (10 mL) to obtain the total mass, but I don’t see why de dilution factor has to be used again. Moreover, I can understand that to refer to membrane weight basis, the total mass corresponding to TAC2 is divided by the weight of the membrane. What it is not clear is why this weight is always 0.4 g exactly. I find it difficult to believe that the authors were able to cut the membrane resulting always in a sample of 0.4 g. My guess is that they always cut a piece of the membrane, and the final weight was near 0.4 g. Equation (4) would be better written using a general term for the weight of the membrane, dividing the total mass extracted. Please check if the DF is meaningful, and if not, please correct the values calculated.
Line 175: equation 8 is not clear, instead of %polymeric color, it should be (polymeric color/color density), so the number is lower than 1 and the result is positive. It is not clear why it has to be multiplied by %polymeric anthocyanins. Please in equation 7 revise spelling.
ANS: Thank you for your comment. Equations 7 and 8 have been modified in the manuscript.
The authors should use the same notation for dilution factor in all the equations. In (3) is DF and in (5) and (6) is “dilution factor”. Still is not clear how multiplying the total monomeric anthocyanins by P and dividing by (K-P) gives the total polymeric anthocyanins in the fouled membrane. Developing equation (8) it is obtained that polymeric anthocyanins(mg/g) = TAC2*P/(K-P).
Lines 268-272: because of the composition of the blueberry extracts mentioned here, it is of highly relevant to provide with the composition of the initial sample and the final concentrate.
ANS: Thank you for your comment. This study focuses on identification of anthocyanins in membrane fouling. The contents of anthocyanins in membrane fouling, extracts and concentrates are shown in Table 1, the chromatogram of anthocyanins is shown in Fig. 4.
The total contents are given in Table 1 (please revise the superscripts that are not defined), but still no reported analysis on the composition of the extracts, which should be indicative of the compounds that are retained/foul the membrane.
Author Response
Dear Editor,
Thank you and the referee for reviewing our revised manuscript (ID: membranes-1134930) carefully. For the comments of reviewer 3#, we discussed and thought about it thoroughly. Please see the details as the following:
- The authors have improved the introduction, adding relevant information regarding previous studies related to their work. In lines 53-54 of the revised manuscript, they have maintained a sentence regarding the need of extracting the foulants for membrane fouling characterization. This is an option, but not the only one. In fact, most of the membrane fouling characterization has been performed based on characterization of foulants deposited on the membrane. The authors should clearly state that extraction is one possibility, but not the only one.
ANS: Thank you for your comment. It has been stated in the manuscript clearly that one of the methods is to extract the compounds on/in the membrane firstly.
- Line 109: “different membranes were tested for their affinity of water at 25ºC” not clear what the authors mean. This is not still clear from my point of view. Do the authors mean that several samples of the NF membrane were tested, or different membranes refer to membranes other than the one employed in the experiments? I think the word “different” can be misleading.
ANS: Thank you for your comment. “Different membranes” should be “pristine membrane and fouled membrane”. It has been revised in the manuscript.
- In Figure 3b it is not possible to clearly read the additions in the figure. Still the quality of the figure could be improved.
ANS: Thank you for your comment. The chromatogram has been modified and improved in the manuscript.
- Thank you for the clarification. I have no problem regarding the method describes in reference [21]. It could be added in the manuscript that the mg/L for TAC1 are referred to cyanidin-3-O-glycoside. My concern is for the way TAC2 is calculated, I see that it has to be multiplied for the volume used in the extraction (10 mL) to obtain the total mass, but I don’t see why de dilution factor has to be used again. Moreover, I can understand that to refer to membrane weight basis, the total mass corresponding to TAC2 is divided by the weight of the membrane. What it is not clear is why this weight is always 0.4 g exactly. I find it difficult to believe that the authors were able to cut the membrane resulting always in a sample of 0.4 g. My guess is that they always cut a piece of the membrane, and the final weight was near 0.4 g. Equation (4) would be better written using a general term for the weight of the membrane, dividing the total mass extracted. Please check if the DF is meaningful, and if not, please correct the values calculated.
ANS: Thank you for your comment. Sorry we did not state it clearly. TAC1 was determined after the dilution of the supernatant obtained from membrane fouling extracts. When TAC1 was determined, DF in the equation was a dilution factor of the buffer solution to the sample. When TAC2 was determined, DF was a dilution factor in the supernatant before the determination of TAC1. In order to avoid ambiguity, the DF in the determination of TAC2 was deleted. Equation (4) has been written using a general term ‘m’ for the weight of membrane.
- The authors should use the same notation for dilution factor in all the equations. In (3) is DF and in (5) and (6) is “dilution factor”. Still is not clear how multiplying the total monomeric anthocyanins by P and dividing by (K-P) gives the total polymeric anthocyanins in the fouled membrane. Developing it is obtained that polymeric anthocyanins(mg/g) = TAC2*P/(K-P).
ANS: Thank you for your comment. The dilution factor in all equations has revised as DF. Equation (8) has modified as:
polymeric anthocyanins(mg/g) = TAC2*P/(K-P)
- Lines 268-272: because of the composition of the blueberry extracts mentioned here, it is of highly relevant to provide with the composition of the initial sample and the final concentrate. The total contents are given in Table 1 (please revise the superscripts that are not defined), but still no reported analysis on the composition of the extracts, which should be indicative of the compounds that are retained/foul the membrane.
ANS: Thank you for your comment. The superscripts have revised. We determined several compounds in the extracts such as total sugar, total pectin, TAC, total phenols, etc, which has been reported in our other manuscript (https://doi.org/10.1016/j.lwt.2021.111196). In this study, we focused on identification of anthocyanins on the fouled membrane. Accordingly, we didn’t mentioned the others. Besides phenols, pectin is the main compound in the extracts.
Please find the revised manuscript in which all the revised terms have been highlighted.
If there is any problem in our current study and manuscript, please feel free to contact me. We will consider it more seriously.
Regards,
All authors

Round 2
Reviewer 1 Report
The authors addressed the comments, the manuscript can be accepted.
Reviewer 2 Report
The authors have introduced appropriate changes. In line 174 of R2 "quality" should be "quantity"??
This manuscript is a resubmission of an earlier submission. The following is a list of the peer review reports and author responses from that submission.
Round 1
Reviewer 1 Report
The manuscript by Cai and co-workers describes the fouling properties of nanofiltration membranes during biophenol extraction from berries. The work is of interest to the membrane community, aligns well with the scope of the journal. The reported research findings have some merit, reasonable novelty, and sufficient data. However, there are several major and minor issues to be addressed prior to further consideration.
1) There is not sufficient information to reproduce the work because the nanofiltration membranes are not disclosed. Section 2.1 mentions the use of membranes but neither manufacturer information nor specifications are given.
2) The chromatographic profiles and mass spectra of anthocyanins in extracts and foulants are not legible. Appropriate resolution should be provided and larger chromatograms are needed because this data cannot be interpreted in any way at the moment.
3) In line 87, specify if permeate flow rate or retentate recirculation rate is reported as 0.08 L/min. In the same paragraph, the effective membrane area should be given as well.
4) The extraction of biophenols with medicinal value such as anthocyanins is an increasingly important topic. Recent works on extraction, in particular exploiting nanofiltration, should be acknowledged (10.1021/acssuschemeng.9b04245; 10.1016/j.foodres.2011.04.046; 10.1016/j.jfoodeng.2016.09.017).
5) In line 80, the authors write that the blueberries were “dissolved” but after the crushing the blueberries won’t dissolve. Most likely the authors prepared a dispersion and not dissolved the crushed fruit. Correct as necessary.
6) The contact angle values are relatively close to each other. Therefore, the authors should report the average (+ error) contact angle of a few independently fouled membranes. In the submitted manuscript, there is no error and therefore the difference in contact angle might not be significant.
7) How long were the fruit extracts stored at -18 degC before use/nanofiltration? This crucial information should be revealed under section 2.2.
8) The applications of biophenols is spreading into different fields with a plethora of potential use in various areas, which should be briefly mentioned (polymer dots 10.1039/D0GC02824J; fuel cells 10.1021/acssuschemeng.0c03058; membranes 10.1021/acsapm.8b00161; hydrogel 10.1039/D0GC02909B).
9) Some error bars are given on performance figures but none of them discusses how the errors were derived. The authors should state in the experimental section if independently obtained data were reported or not, i.e. extracts from blueberry were prepared separately and then subjected to nanofiltration, or the same fouled membranes were analysed repeatedly. Preferably the former one should be reported.
10) The conclusion section should summarize the main research findings in quantitative statements as well.
11) Both the quotient (“x/y”) and negative exponent (“x y-1”) formats are used in the manuscript for units. Either of them should be used consistently, preferably the negative exponent format, which is recommended by the IUPAC.
12) The introduction discusses only 8 literature references, which is insufficient to provide an appropriate background and context to all aspects of the research detailed in the manuscript.
13) Provide purity information for all chemicals used in the study under section 2.1.
Reviewer 2 Report
The manuscript “Identification of anthocyanins and their fouling mechanisms during non-thermal nanofiltration of blueberry aqueous extracts” studies the anthocyanins retained by a nanofiltration membrane used to concentrate a blueberry extract. Even though the authors show an extensive effort in the chemical characterization of the compounds retained by the membrane during de NF process, I have doubts about how deep they go to study membrane fouling that can be applied to improve the concentration process or to design cleaning protocols for the membrane. To achieve that, information on the characteristics of the blueberry extract and concentrate would be required, as well as an experimental plan including different NF conditions, that as far as I could understand have not been performed. Moreover, there are several issues that the authors should amend before the manuscript is published, such as:
Lines 58-60:
The authors refer to membrane fouling during wine clarification, which is most probably carried out by microfiltration. Fouling mechanisms for nanofiltration should be referred. It is true that even though the size of the molecules is much smaller than the microfiltration pores, fouling occurs due to the interactions between these molecules (polyphenols) and other molecules present. But there is a scarce review on polyphenol fouling during nanofiltration in the introduction.
Section 2.3: The authors mention that NF was performed at 10 bar, did they test other pressures or velocities? As for the NF membrane, the only information given consists only of the molecular weight cut-off (section 2.1). Was it a tubular membrane? A flat membrane? Commercial? Custom-made? There is no information about the NF equipment: a schematic could be helpful to understand about how the permeate and retentate were collected and measured. There is no information about replicates, even though figure 1a shows error bars. There is also a very important step in NF which is membrane conditioning. The authors only refer to a method to remove the preserving agents, but no filtration of water through the membrane was performed. Since membrane compaction would affect its performance, the authors should clearly state if conditioning of the membrane filtering water at a pressure close to the one used for concentration was performed.
Line 109: “different membranes were tested for their affinity of water at 25ºC” not clear what the authors mean.
In Figure 3b it is not possible to clearly read the additions in the figure.
Lines 123-125: the extraction of the fouling compounds ends with centrifugation of the extract to remove the residue. Was the residue part of the fouling material? Could it be that some aggregates were formed and are removed and no analyzed? This is also related with the lack data on the characterization of the extract and concentrate.
Lines 144-159: it is not clear how they perform the determination of the TAC2. First, they obtain the mg/mL of monomeric anthocyanins in the extracts from the membrane (TAC1). Then, they calculate the monomeric anthocyanins for the fouled membrane, using the previous value in mg/mL, which is multiplied by a dilution factor (that was already used in the TAC1) and divide by the weight of membrane (quantity not quality). Therefore, the final result is referred to the g of membrane. Why do they use these units? Would not be better to refer to membrane surface unit? Also, in this part it should be clarified if there is any difference between the determinations in the extracts and in the fouled membrane.
Line 175: equation 8 is not clear, instead of %polymeric color, it should be (polymeric color/color density), so the number is lower than 1 and the result is positive. It is not clear why it has to be multiplied by %polymeric anthocyanins. Please in equation 7 revise spelling.
Lines 268-272: because of the composition of the blueberry extracts mentioned here, it is of highly relevant to provide with the composition of the initial sample and the final concentrate.
Reviewer 3 Report
In this work, Authors found that amphiphilic molecules anthocyanins in blueberry extracts are a potential foulant during its NF concentration. Therefore, the membrane contact angle was increased from the initial value of 30.3o to 42.3o after concentration, means that the hydrophobicity of membrane was increased. Even though it is not indicating a significant change of hydrophobicity, the information regarding contact of this such foulant on the membrane can be used for NF membrane development for the prevention and control of fouling, for the choice of specific agents in cleaning solutions for NF membrane. This work is interesting, and then I am able to recommend this work to be published in the MDPI membranes journal.